# The Effect of Chlorogenic Acid on *Bacillus subtilis* Based on Metabolomics

**DOI:** 10.3390/molecules25184038

**Published:** 2020-09-04

**Authors:** Yan Wu, Shan Liang, Min Zhang, Zhenhua Wang, Ziyuan Wang, Xin Ren

**Affiliations:** 1Beijing Advanced Innovation Center for Food Nutrition and Human Health, Beijing Technology and Business University, Beijing 100048, China; 10011216006@st.btbu.edu.cn (Y.W.); liangshan@btbu.edu.cn (S.L.); zhwang@btbu.edu.cn (Z.W.); wangziyuan@btbu.edu.cn (Z.W.); 20180734@btbu.edu.cn (X.R.); 2Beijing Engineering and Technology Research Center of Food Additives, Beijing Technology and Business University, Beijing 100048, China

**Keywords:** chlorogenic acid, *Bacillus subtilis*, antimicrobial activity, metabolomics

## Abstract

Chlorogenic acid (CGA), a natural phenolic compound, is an important bioactive compound, and its antibacterial activity has been widely concerned, but its antibacterial mechanism remains largely unknown. Protein leakage and the solution exosmosis conductivity of *Bacillus subtilis 24434* (*B. subtilis*) reportedly display no noticeable differences before and after CGA treatment. The bacterial cells treated with CGA displayed a consistently smooth surface under the electron microscope, indicating that CGA cannot directly disrupt bacterial membranes. However, CGA induced a significant decrease in the intracellular adenosine triphosphate (ATP) concentration, possibly by affecting the material and energy metabolism or cell-signaling transduction. Furthermore, metabolomic results indicated that CGA stress had a bacteriostatic effect by inducing the intracellular metabolic imbalance of the tricarboxylic acid (TCA) cycle and glycolysis, leading to metabolic disorder and death of *B. subtilis*. These findings improve the understanding of the complex action mechanisms of CGA antimicrobial activity and provide theoretical support for the application of CGA as a natural antibacterial agent.

## 1. Introduction

Chlorogenic acid (CGA) is an important phenolic compound belonging to the hydroxycinnamic acid derivative family and consists of caffeic acid and quinic acid via esterification [1,2]. CGA is abundantly present in coffee beans [3], fruits [4], vegetables [5], and herbs [6] and displays antibacterial, antiphlogistic, antimutagenic, antioxidant, and other biological activities [7,8]. Studies are directing increasing focus toward the extraction, separation, and antibacterial activity of CGA in a variety of plants [9], indicating that this compound is effective against certain microorganisms [10]. Kim and Li et al. [11,12] confirmed that CGA displayed an antimicrobial effect, while Li et al. [12] indicated that the minimum inhibitory concentration (MIC) of CGA against *Staphylococcus aureus* was 2.5~5.0 mg/mL. Sung et al. [5] concluded that CGA exhibited anti-fungal activity, while the research conducted by Wang et al. [13] involving the antibacterial effect of tobacco CGA extract showed a certain inhibitory effect on *Escherichia coli* and *Bacillus subtilis*. Lou et al. [14] reported that the MIC for CGA against *B. subtilis* was 40 μg/mL. Su et al. [15] indicated that the MIC of CGA against *Pseudomonas fluorescein* and *Staphylococcus saprophytes* from chicken was 5 mg/mL. Currently, it is speculated that the way in which the structure and function of CGA affects its antimicrobial activity occurs as follows: (1) CGA molecule contains phenolic hydroxyl, which can affect the activity of related metabolic enzymes, reduce the metabolic level, cause the metabolic process to be blocked, and inhibit the activity of bacteria [16]. (2) CGA molecules display strong polarity and can bind to large molecules, such as lipids, on the surface of bacteria to change the permeability of the bacterial membrane, resulting in the leakage of the contents [14,17]. (3) CGA may inhibit bacterial flagella synthesis, reduce the number of flagella, and thus reduce the bacterial clustering effect [18].

Food spoilage, in a broad sense, is a change in the chemical or physical properties of food caused by microorganisms, which poses a serious threat to food safety [19]. However, the reproduction of microorganisms on both the exterior and interior of food is the most common cause of food spoilage [20]. Food spoilage *Bacillales* members are typically assigned to the *Bacillus, Geobacillus, Anoxybacillus, Alicyclobacillus*, and *Paenibacillus genera* [21]. The Gram-positive bacteria responsible for spoilage in meats are *Clostridium*, *Bacillus*, and *Lactic acid bacteria* [22]. Xu et al. [23] isolated and identified the dominant bacteria causing fresh noodle corruption and found that the dominant strains causing instant wet surface deterioration were mainly *Bacillus subtilis, Bacillus licheniformis,* and *Bacillus giganteum*. Li et al. [24] found that *Bacillus licheniformis* and *Bacillus subtilis* were the dominant bacterial strains during the early stages of storage. Therefore, according to previous studies, *Bacillus subtilis* displays a strong ability to hydrolyze starch and protein, while this microorganism commonly causes food spoilage.

The metabolome, which refers to the systematic analysis of the small molecule metabolites in microbial compositions and dynamic responses [25], encompasses a diverse array of molecular chemotypes, including peptides, carbohydrates, lipids, nucleosides, and catabolic products of exogenous compounds [26]. The metabonomic analysis is used to assess the series of changes occurring under external stress or pathological damage [27], while metabolites directly reveal a life of phenotypic change in the system [28].

Therefore, the study of the changes in intracellular metabolites assists in understanding the effect of CGA on microbial metabolism, attempting to elucidate the antibacterial activity of CGA further. Accordingly, this study clarifies the effect of CGA on *B. subtilis* by examining the effect of bacteriostatic agents at different concentrations on bacterial metabolites, while conducting an extensive investigation into the bacteriostatic activity. Consequently, this study provides a theoretical basis for the development and utilization of CGA bacteriostatic agents.

## 2. Results and Discussion

### 2.1. MIC

In this study, the MIC values for CGA was 2.50 mg/mL. However, Lou et al. [14] reported that the MIC for CGA against *B. subtilis* was 40 μg/mL, which was lower than the value calculated in this study. Su et al. [15] indicated that the MIC of CGA against *Pseudomonas fluorescein* and *Staphylococcus saprophytes* was 5 mg/mL. A possible reason is that different CGA extraction conditions and variations in the inoculum level, experimental temperature, and physiological condition of the bacteria used in different studies may affect its antibacterial activity.

### 2.2. Scanning Electron Microscope (SEM) and Membrane Permeability Assay

Any morphological changes in the tested strains treated with CGA were observed with SEM to observe the damage to the cell structure. Figure 1A shows that the untreated bacteria exhibited bacilliform morphology, and the surface appeared plump and intact. The bacterial cells treated with CGA at MIC values were slightly damaged compared with the control and 0.5 MIC CGA cells treated, which had a smooth surface. The release of intracellular proteins and changes in conductivity can be regarded as an indication of cell structure integrality [29]. Figure 1B shows that there was no significant change in protein leakage before and after CGA treatment. The effect on the cell membrane in response to CGA treatment was also observed through relative permeability (Figure 1C). The relative conductivity of *B. subtilis* did not increase substantially after CGA treatment, while displaying no significant differences from the control group. The results showed that CGA treatment had a negligible effect on the cell membrane of *B. subtilis*.

ATP represents the direct energy source in organisms. Figure 1D shows the effect of different CGA concentrations on the intracellular ATP levels in the *B. subtilis* cells, which was 420.01 μmol/mL in the control group. In comparison, *B. subtilis* treated MIC CGA showed a significant (*p* < 0.05) reduction in the intracellular ATP concentrations, while exposure to MIC CGA produced a value of 358.49 μmol/mL in ATP levels. Compared with MIC CGA, 0.5 MIC CGA induced a decrease in the intracellular ATP concentrations, which was lower than CGA treatment alone. Since the cell membrane was not damaged, the decrease in ATP was probably caused by the influence of CGA on the intracellular metabolism or intracellular signal transduction. Therefore, the changes in intracellular metabolism were detected after CGA treatment.

### 2.3. The Intracellular Metabolites of B. subtilis

A total of 81 intracellular metabolites with known structures were identified in *B. subtilis* after CGA treatment at different concentrations (Table 1). These included amino acids, organic acids, phosphoric compounds, cofactors, and nucleotides, as well as hormones and secondary metabolites. The highest number of intracellular metabolites detected in this study were also reported in the metabolomics studies conducted by Ding et al. [27] and Bo et al. [30].

Principal component analysis (PCA), an unsupervised clustering method, was used to process the data to verify the observations further and identify the metabolites primarily responsible for the discrimination between the cells treated with CGA and the control cells. The results indicated that the cumulative contribution rate of the first seven principal components was 81.62%, which reflected the primary variable data. As shown in Figure 2, the score plots of both the first principal component (P1) and the second principal component (P2) depicted a clustering of samples. In the PCA score chart (R2X = 0.504), the horizontal and vertical coordinates were P1 and P2, while the contribution rates of these two principal components were 0.397 and 0.107, respectively. The CGA-treated samples were distinctly different from the control group. The results suggested that CGA had a noticeable effect on cell metabolism, which again confirmed that the inhibition of *B. subtilis* by CGA might be achieved by restricting the metabolism.

Hierarchical cluster analysis (HCA) was employed to perform a preliminary assessment of the samples, and the obtained heatmap (Figure 3) not only reflected the differences and similarities between the various samples but also showed the content diversity of the 81 metabolites following the addition of CGA. The compositional differences between the concentrations were much more pronounced during HCA. The heatmap indicated that the addition of CGA led to significant differences between each experimental group and the control group, while differences were evident in the *B. subtilis* cell metabolic map before and after CGA treatment. Compared with the control group, the level of intracellular metabolites declined after the addition of CGA. The content of 37 metabolites decreased significantly after CGA treatment, which included citric acid, malonic acid, and glycerol 3-phosphate, among others. These metabolites affected several metabolic pathways, such as glycolysis, tricarboxylic acid (TCA) metabolism, amino acid metabolism, pentose phosphate, and pyrimidine metabolism, while increasing the orotic acid and putrescine content. Putrescine is a regulatory metabolic substance that can interact with negatively charged nucleic acids, membrane proteins, and other biological macromolecules, resulting in a series of physiological or pathological changes in the cell. This study observed that compared with the control group, the addition of CGA induced either an increase or a decrease in the bacterial metabolite content, leading to the metabolic disorder and death of bacteria. Based on the variation in metabolite concentration, combined with the experimental results of CGA regarding membrane integrity, CGA at 0.5 MIC level had no significant effect on membrane integrity but had a considerable impact on intracellular metabolite concentration. Therefore, CGA can inhibit bacterial growth and even cause death by affecting the metabolic pathways.

The metabolites of the individual CGA-treated and control groups were clustered and could be discriminated from each other with the multivariate statistical method, orthogonal partial least squares discriminant analysis (OPLS-DA) [31]. After cross-validation via pali-pair comparison, the differential metabolites were screened using a load graph, while a shared and unique structure (SUS)-plot model was adopted for screening these metabolites among the three groups. Furthermore, the differences in the metabolites of the *B. subtilis* concentrations after CGA treatment were also determined (Figure 4). From the results, most of the intracellular metabolites changes with CGA concentration and can be divided into glycolysis, purine metabolism, amino acids, and carbohydrates according to the metabolic pathway or function. Furthermore, it is believed that CGA treatment can block the transformation of key metabolites in the pathway and cause the metabolic imbalance of small cell molecules, therefore, affecting bacterial activity. Halouska and Fenton et al. [32] employed OPLS-DA to profile the in vivo action mechanism of known antibiotics used to treat *M. tuberculosis*.

### 2.4. The Effect of CGA on the Primary Metabolism of B. subtilis

The measured metabolite variations were mapped onto the metabolic pathways (Figure 5) to investigate potential links between the metabolic changes and CGA treatment. The TCA cycle forms a crucial part of the metabolic pathways of all aerobic organisms during the generation of energy [33]. Compared with the experimental group, the citrate, cis-aconitate, isocitrate, succinate, and other substances changed significantly during the TCA cycle. During glycolysis, glucose 6-phosphate, and fructose 1, 6-diphosphate were significantly reduced in the CGA-treated group compared with the control group. Citrate, cis-aconitate, and succinate are related to the TCA cycle. After 2 h, the citrate and cis-aconitate levels decreased dramatically in the CGA-treated group compared with the control group, while the succinate levels showed a marked decrease in the 0.5 MIC and MIC CGA-treated groups, respectively. Tao et al. [30] suggested that the cause of the pathway changes after CGA treatment might be related to the inhibition of the metabolic level of the *B. subtilis* TCA cycle by CGA. Therefore, CGA reportedly affected the bacterial metabolism, resulting in insufficient energy for the bacteria, while impacting their growth and reproduction [34].

Changes in amino acids, which play a critical role in cells and participate in a variety of life activities, may affect the normal functionality of cells. We show examples of 15 amino acids that differed significantly between the three groups in Figure 6, including L-Lysine (Lys), L-Methionine (Met), L-Phenylalanine (Phe), L-Proline (Pro), L-Serine (Ser), L-Tyrosine (Tyr), L-Alanine (Ala), L-Arginine (Arg), L-Aspartic acid (Asp), L-Glutamate (Glu), L-Histidine (His), and L-Isoleucine (Ile). Met, Phe, Tyr, Asp, Glu, and Ile were substantially higher in the control group than in CGA-treated groups. Glu is associated with the carbon and nitrogen balance of cells, while the synthesis of nitrogen compounds in cells generally requires glutamate to provide a nitrogen source [35]. Higher levels of asparagine often correlate with differences in the carbon/nitrogen balance. Metabolites, including Ser, phenylalanine, Leu, Ile, and Val, can be converted from 3-P-glycerate and pyruvate, which are both metabolic intermediates of the Embden-Meyerhof-Parnas (EMP) pathway [30]. In previous studies, the *S. cerevisiae* EMP pathway was also inhibited by phenol, acetic acid, and other external stimulants [27,36] as the levels of the metabolic intermediates of the EMP pathway decreased under CGA treatment. These results also indicated the inhibition of the EMP pathway by CGA.

## 3. Materials and Methods

### 3.1. The Reagent and Bacterial Strains

*B. subtilis* was obtained from the China Center of Industrial Culture Collection (CICC). Luria-Bertani (LB) medium (5 g beef extract, 10 g peptone, 10 g NaCl, and 1000 mL H_2_O, at pH 7.2). A stock solution of 10.0 mg/mL CGA (95% purity, Shanghai Yuanye Bio-Technology Co., Ltd., Shanghai, China) was prepared.

### 3.2. Determination of MIC

The MIC of the antimicrobials was determined using the standard broth microdilution method [37], with some modifications. *B. subtilis* was incubated in LB medium at 37 °C for 8–10 h to reach approximately 10^6^ CFU/mL. Serial dilutions of CGA were prepared in an LB medium to obtain final concentrations of 5 mg/mL, 2.5 mg/mL, 1.25 mg/mL, 0.625 mg/mL, 0.3125 mg/mL, and 0.15625 mg/mL. The plates were incubated for up to 24 h before recording the MICs. Samples incubated without antimicrobials were used as controls.

### 3.3. The Detection of Extracellular Protein

The activated indicator bacteria were inoculated into an LB medium at a volume concentration of 2% and cultured to the logarithmic phase after which it was centrifuged and resuspended in phosphate-buffered saline (PBS) buffer at pH 7.4. CGA at a concentration of 1 × MIC was added to the culture for 2 h and centrifuged after which the supernatant was collected for use. The Bradford method was used to determine the absorbance at 595 nm, while the protein concentration was calculated according to the protein standard curve and the sample volume. The experiment was repeated three times with the indicator bacteria and PBS (pH 7.4) buffer as the control group. Please refer to the instruction manual of the test kit for determining the mass protein concentration.

### 3.4. SEM Assay

SEM was used to observe the morphological changes in *B. subtilis* after CGA treatment with 0.5 MIC and MIC. Logarithmic phase bacteria were exposed to different concentrations of CGA for 2, 4, and 8 h after which the cells were washed with PBS following incubation and fixed for 1 h at 4 °C with 2.5% glutaraldehyde. The samples were dehydrated in sequential ethanol, freeze-dried with a vacuum freeze dryer, coated using an ion sputtering apparatus (Hitachi MC 10,003), and observed with SEM (Hitachi SU8020, Hitachi Productions Inc., Tokyo, Japan). The bacterial cells that were not treated with the compounds were similarly processed and served as controls. High-powered micrographs from multiple distinct low-powered fields in each sample were obtained for the quantitation of damaged cells. Cells that had lost their original shapes and displayed a smooth cell wall were positively scored for damage, such as wrinkling, distortion, and lysis.

### 3.5. Membrane Permeability Assay

The activated bacteria were transferred into LB medium and cultured at 37 °C until the logarithmic stage (~10^6^ CFU/mL). Respective concentrations of 0.5 MIC and MIC CGA were added to sterile centrifuge tubes after which three parallel samples were collected from each concentration without CGA as a blank control. Furthermore, 0.2 mL of bacterial solution was taken from each centrifuge tube and mixed well. The initial conductivity in the centrifuge tube was measured as C_0_ using a conductivity meter after which the solution was cultured at 37 °C, and removed after 2, 4, and 8 h, respectively. Group A was centrifuged at room temperature for 5 min at 12,000 rpm, while group B was bathed in water at 100 °C for 30 min, cooled to room temperature at 12,000 rpm, centrifuged at room temperature for 5 min after which the supernatant was collected. The conductivity of both group A and group B was determined using a conductivity meter, which was denoted as C_A_ and C_B_, respectively. The measured conductivity was calculated with a formula where the relative permeability and the permeability of the cell membrane were compared:$$\mathrm{Relative}\;\mathrm{permeability}=\frac{C_{A}-C_{0}}{C_{B}-C_{0}}\times 100\%$$

### 3.6. Measurement of the ATP Content

The method described by Turgis et al. [38] was followed with some modifications. The working culture of *B. subtilis* was centrifuged for 5 min at 12,000 rpm, and the supernatant was removed. The cell pellets were washed three times with PBS (pH 7.4) after which the cells were collected via centrifugation. A cell suspension (OD600 = 0.3~0.4) was prepared with 50 mL PBS (pH 7.4) after which 4 mL of the cell solution was placed into an Eppendorf tube for treatment with 0 mg/mL (control), 0.5 MIC CGA, and MIC CGA, respectively. Samples were maintained at 37 °C for 2, 4, and 8 h, respectively, and centrifuged for 5 min at 12,000 rpm. Then the cells were left on ice to prevent ATP loss until measurement of the intracellular ATP concentrations occurred using an ATP assay kit (Beyotime Institute of Biotechnology, Shanghai, China). The ATP cell concentration, which represented the intracellular concentration, was determined using a spectrophotometer (Thermo Scientific™ Varioskan™ LUX, Thermo Fisher Scientific Inc., Waltham, MA, USA), following the instructions of the manufacturer.

### 3.7. Extraction of the Intracellular Metabolites

*B. subtilis* was precultured in LB liquid medium with a rotary speed of 150 rpm at 37 °C for 12 h. Then, the precultured bacteria cells were transferred to a fresh LB liquid medium, and the initial cell density was adjusted to 0.3–0.4 at 600 nm. The cells were cultivated at 37 °C on a rotary shaker at 150 rpm in 250 mL cotton-plugged flasks containing 100 mL of LB liquid medium either with or without CGA. Cell samples were collected at 2, 4, and 8 h either with or without CGA. First, bacteria cells were harvested by centrifugation at 12,000 rpm for 5 min and washed with PBS three times to remove the residual culture medium, followed by washing with ultrapure water to remove the salts from PBS. Then, we put the cells on dry ice and add 2 mL of 80% (*v*/*v*) methanol (pre-chilled to −80 °C). Then, cells were broken the cells using sonication under 10 °C and incubated the lysate at −80 °C for 2 h. Then, the metabolite-containing samples were harvested by centrifugation at 12,000 rpm for 10 min at 4–8 °C, and we collected the metabolite-containing supernatant to a new 1.5-mL tube and prepared for analyses. Three replicates were performed for each sample.

This experiment was analyzed using TSQ Quantiva (Thermo, Waltham, CA, USA). Samples were separated using a Synergi Hydro-RP column (2.0 × 100 mm, 2.5 μm, Phenomenex, Torrance, CA, USA). A binary solvent system (mobile phase A, 10 mM tributylamine adjusted with 15 mM acetic acid in water; mobile phase B, methanol) was used. This analysis focused on the TCA cycle, the glycolysis pathway, the pentose phosphate pathway, amino acids, and purine metabolism. A 25 min gradient with a flow rate of 250 μL/min was applied as follows: 1–5 min at 5% B; 5.1–20 min, 5–90% B; 20.1–25 min, 90% B. Positive-negative ion switching mode was performed for data acquisition. The resolution for Q1 and Q3 were both 0.7 FWHM. The source voltage was 3500 v for the positive and 2500 v for the negative ion mode. The source parameters were as follows: spray voltage: 3000 v; capillary temperature: 320 °C; heater temperature: 300 °C; sheath gas flow rate: 35; auxiliary gas flow rate: 10. Data analysis and quantitation were performed using the TraceFinder 3.2 software (Thermo Fisher, Waltham, CA, USA).

An in-house database for endogenous metabolites identification was created using Library Manager 2.0 (Thermo Fisher Scientific, CA, USA). Most reference spectra in the internal metabolite library were acquired from chemical standards. In some cases, standard metabolites were not accessible but were observed in biological samples. MS/MS spectra of these compounds were confirmed manually according to Metlin (www.metlin.scripps.edu) or HMDB (www.HMDB.ca) and also saved in Library Manager for reference. All of the areas of acquired peaks were normalized against the internal standard for further data processing.

### 3.8. Statistical Analysis

All experiments were performed at least three times to obtain the value denoting the average ± standard deviation (SD). All the areas of acquired peaks were normalized against the internal standard for further data processing. These normalized peak areas (variables) were imported into SIMCA (ver. 14) (Umetrics, Umeå, Sweden) for multivariate statistical analysis. Principal component analysis (PCA) and orthogonal partial least squares discriminant analysis (OPLS-DA) was applied to the data after mean-centering and orthogonal signal correction preprocessing. Moreover, another unsupervised hierarchical cluster analysis (HCA) was performed using the MeV software (4.8). T-tests (SPSS 22) were performed on specific metabolites and their ratios to assess the statistical significance of the metabolic changes.

## 4. Conclusions

In summary, the SEM results indicate that *B. subtilis* cells can remain morphologically almost unchanged after exposure to CGA. The results of the membrane permeability assay, including the leakage of proteins and the solution exosmosis conductivity, indicate that no significant difference is evident after CGA treatment of *B. subtilis* cells. Furthermore, metabolomics analysis reveals that CGA stress leads to the inhibition of metabolic pathways through the suppression of the TCA cycle and glycolysis. Therefore, it is likely that the bacteriostatic action of CGA on *B. subtilis* may be achieved by inducing intracellular metabolic imbalance.

## Figures and Tables

**Figure 1 molecules-25-04038-f001:**
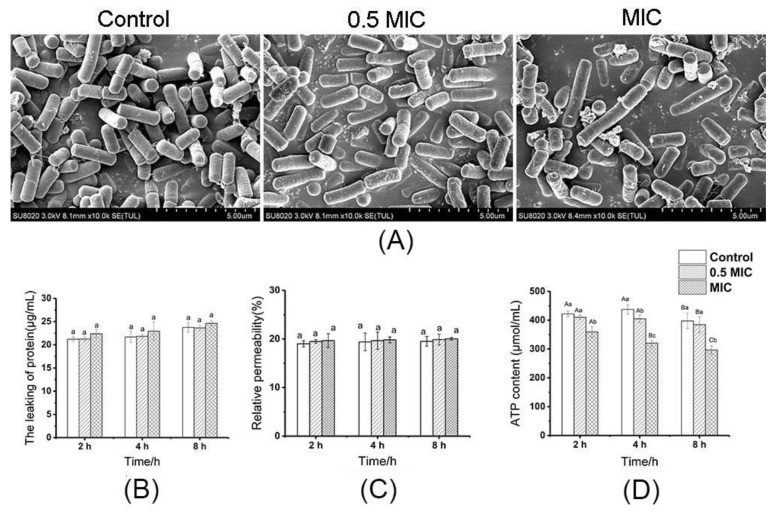
Effect of different chlorogenic acid (CGA) concentrations on *B. subtilis*: (**A**) SEM images; (**B**) protein leakage; (**C**) the change in relative permeability; (**D**) intracellular ATP content. The values represent the means of three reproducible experiments. (Capital letters indicate the significance of differences at different times, while lowercase letters indicate the significance of differences between samples at the same time.).

**Figure 2 molecules-25-04038-f002:**
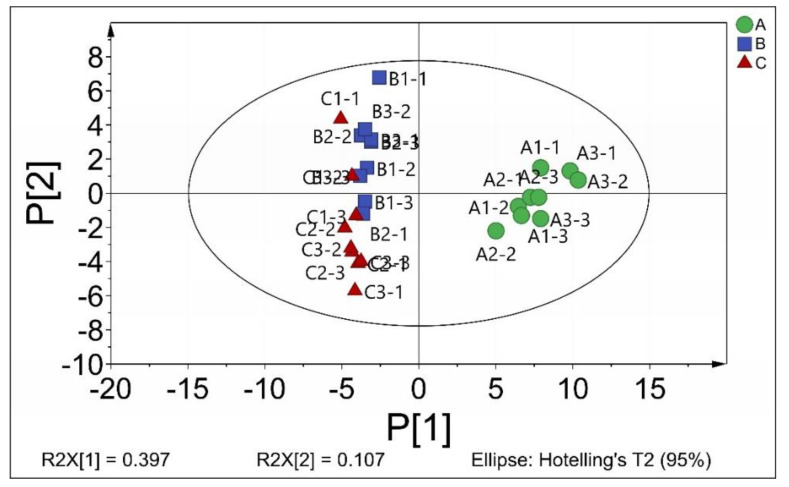
Principal component analysis (PCA) of the effect of different CGA concentrations on the metabolites of *B. subtilis*: A (Control); B (0.5 MIC); C (MIC). MIC = minimum inhibitory concentration.

**Figure 3 molecules-25-04038-f003:**
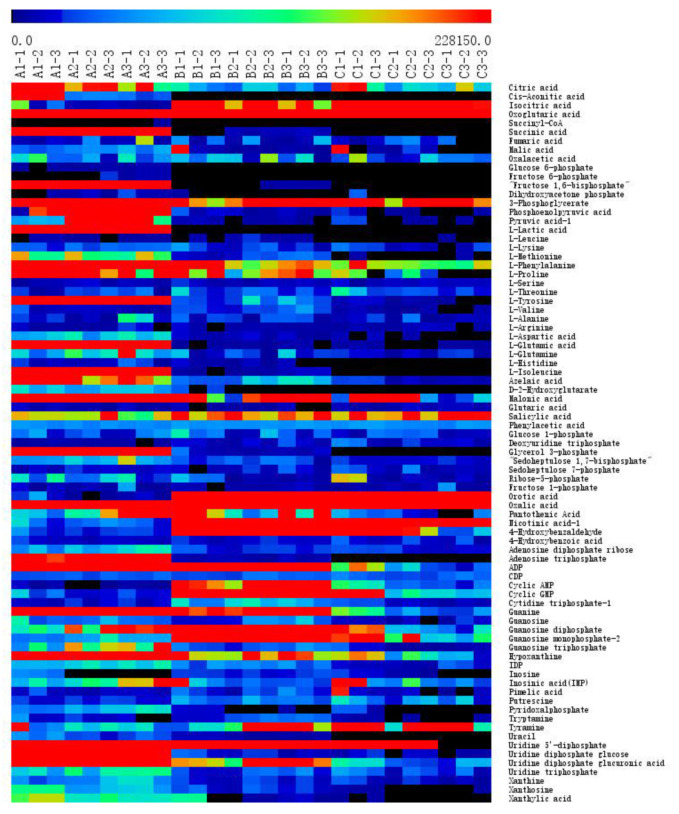
Heat map representation of the effect of different CGA concentrations on the *B. subtilis* metabolites. A: Control group; B: 0.5 MIC-treated group; C: MIC-treated group: A (Control); B (0.5 MIC); C (MIC).

**Figure 4 molecules-25-04038-f004:**
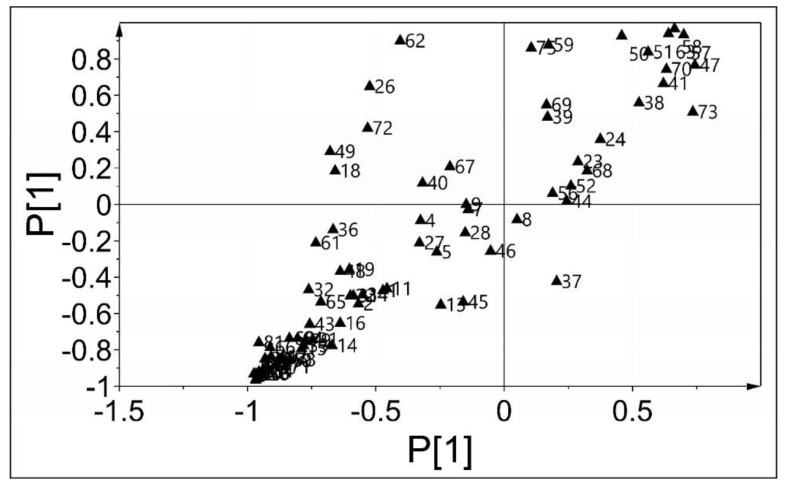
SUS-plot differentiating the effect of various CGA concentrations on *B. subtilis* metabolites.

**Figure 5 molecules-25-04038-f005:**
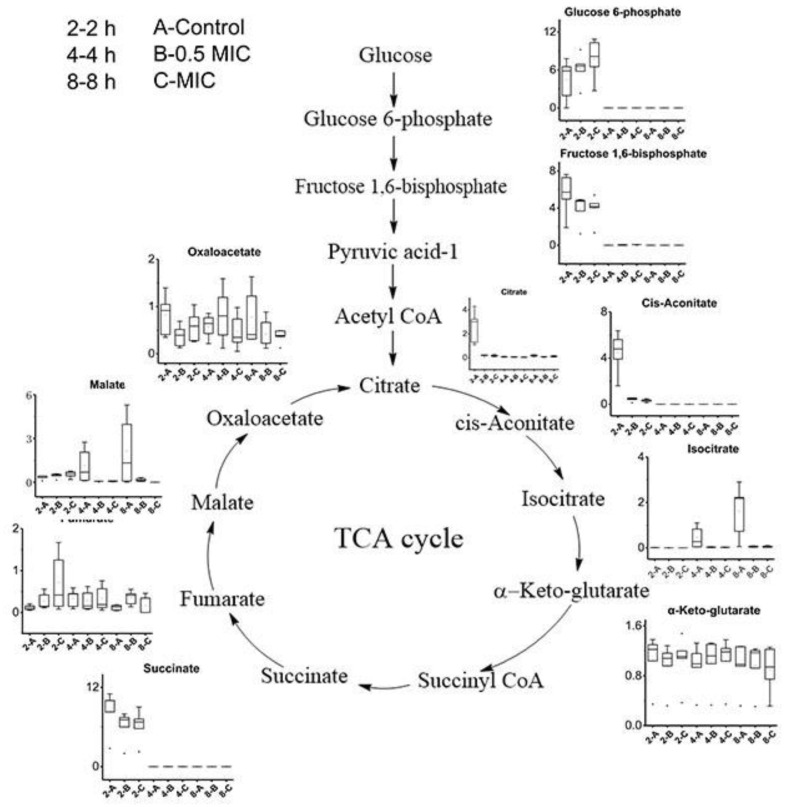
Schematic showing changes in metabolite abundance, which was mapped using the main metabolic network at sampling times of 2, 4, and 8 h.

**Figure 6 molecules-25-04038-f006:**
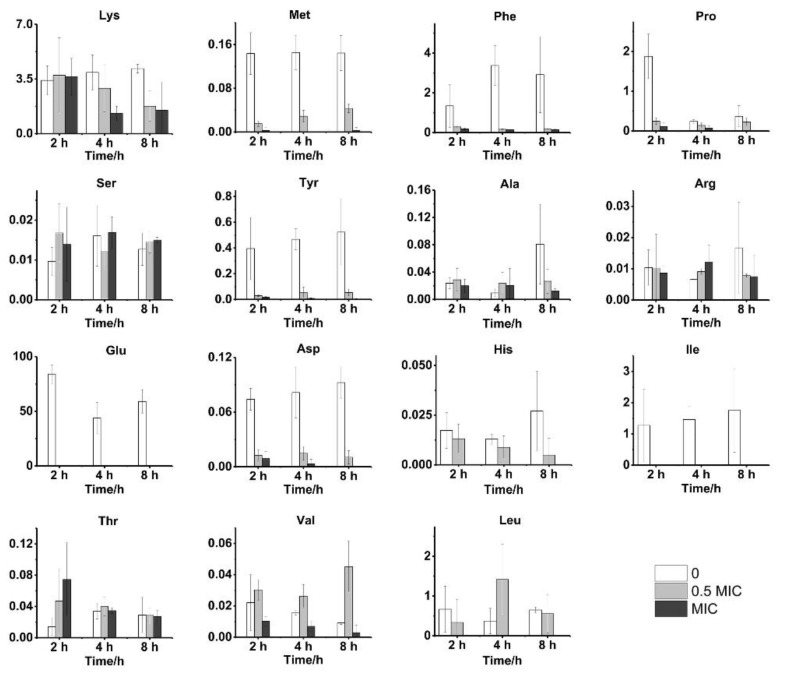
Amino acid variations under the influence of CGA at 2, 4, and 8 h.

**Table 1 molecules-25-04038-t001:** Intracellular metabolites in *B. subtilis* before and after CGA treatment as detected with LC-MS.

NO.	Compound Name	NO.	Compound Name	NO.	Compound Name
1	Citric acid	28	L-Arginine	55	ADP
2	Cis-Aconitic acid/suberic acid	29	L-Aspartic acid	56	CDP
3	Isocitric acid	30	L-Glutamic acid	57	Cyclic AMP
4	Oxoglutaric acid	31	L-Glutamine	58	Cyclic GMP
5	Succinyl-CoA	32	L-Histidine	59	Cytidine triphosphate-1
6	Succinic acid	33	L-Isoleucine	60	Guanine
7	Fumaric acid	34	Azelaic acid	61	Guanosine
8	Malic acid	35	D-2-Hydroxyglutarate	62	Guanosine diphosphate
9	Oxalacetic acid	36	Malonic acid	63	Guanosine monophosphate-2
10	Glucose 6-phosphate	37	Glutaric acid	64	Guanosine triphosphate
11	Fructose 6-phosphate	38	Salicylic acid	65	Hypoxanthine
12	Fructose 1,6-bisphosphate	39	Phenylacetic acid	66	IDP
13	Dihydroxyacetone phosphate	40	Glucose 1-phosphate	67	Inosine
14	3-Phosphoglycerate	41	Deoxyuridine triphosphate	68	Inosinic acid(IMP)
15	Phosphoenolpyruvic acid	42	Glycerol 3-phosphate	69	Pimelic acid/2-oxoadipate
16	Pyruvic acid-1	43	Sedoheptulose 1,7-bisphosphate	70	Putrescine
17	L-Lactic acid	44	Sedoheptulose 7-phosphate	71	Pyridoxal phosphate
18	L-Leucine	45	Ribose-5-phosphate	72	Tryptamine
19	L-Lysine	46	Fructose 1-phosphate	73	Tyramine
20	L-Methionine	47	Orotic acid	74	Uracil
21	L-Phenylalanine	48	Oxalic acid	75	Uridine 5′-diphosphate
22	L-Proline	49	Pantothenic Acid	76	Uridine diphosphate glucose
23	L-Serine	50	Nicotinic acid-1	77	Uridine diphosphate glucuronic acid
24	L-Threonine	51	4-Hydroxybenzaldehyde	78	Uridine triphosphate
25	L-Tyrosine	52	4-Hydroxybenzoic acid	79	Xanthine
26	L-Valine	53	Adenosine diphosphate ribose	80	Xanthosine
27	L-Alanine	54	Adenosine triphosphate	81	Xanthylic acid

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
