# Peer review of "The Effect of Chlorogenic Acid on Bacillus subtilis Based on Metabolomics"

_molecules, 2020, doi:10.3390/molecules25184038_

Round 1
Reviewer 1 Report
Dear authors,
Yan Wu , Shan Liang , Min Zhang, Zhenhua Wang , Ziyuan Wang , Xin Ren,
your article
The Effect of Chlorogenic Acid on Bacillus Subtilis based on Metabolomics
MS molecules-917061 is well prepared and could be accepted as suitable for Molecules readers.You have found that chlorogenic acid (CGA) stress had a bacteriostatic effect by inducing the intracellular energetic metabolic imbalance of the tricarboxylic acid (TCA) cycle and glycolysis, leading to metabolic disorder and death of Bacillus subtilis 24434 (B. subtilis). These findings improve the understanding of the complex action mechanisms of CGA antimicrobial activity and provide theoretical support for the application of CGA as a natural antibacterial agent. You have detected 81 intracellular metabolites which content was estimated. Content of 37 metabolites decreased by CGA treatment. CGA stress have markedly changed TCA cycle metabolites, as well amino acid levels and mutual content proportions. Principal component analysis (PCA) and hierarchial cluster analysis (HCA) were done. Please check some errors: P.1, line 37, line 38 and further in all the text flow use et al. in Italics. Line 90 (similarly in line 162) - insert coma after Figure 1, III. In Conclusions, lines 288, 291 and 293 use B. subtilis in Italics.
Author Response
Response to Reviewer 1 Comments
Point: You have found that chlorogenic acid (CGA) stress had a bacteriostatic effect by inducing the intracellular energetic metabolic imbalance of the tricarboxylic acid (TCA) cycle and glycolysis, leading to metabolic disorder and death of Bacillus subtilis 24434 (B. subtilis). These findings improve the understanding of the complex action mechanisms of CGA antimicrobial activity and provide theoretical support for the application of CGA as a natural antibacterial agent. You have detected 81 intracellular metabolites which content was estimated. Content of 37 metabolites decreased by CGA treatment. CGA stress have markedly changed TCA cycle metabolites, as well amino acid levels and mutual content proportions. Principal component analysis (PCA) and hierarchial cluster analysis (HCA) were done. Please check some errors: P.1, line 37, line 38 and further in all the text flow use et al. in Italics. Line 90 (similarly in line 162) - insert coma after Figure 1, III. In Conclusions, lines 288, 291 and 293 use B. subtilis in Italics.
Response: Thanks for the comment. We have revised all the text “et al.” in Italics. We have inserted coma after Figure 1, III (Line 94). We have revised “B. subtilis” in Italics. We have revised these errors in the text, which we have marked with the track changes mode.
Reviewer 2 Report
This manuscript details a study on the metabolomic changes in Bacillus subtilis when subjected to high concentrations of chlorogenic acid. The work is scientifically sound, and the statistical analyses were generally well done. The topic is a little underwhelming, as the MIC values reported in the literature and this study are well into the mg/ml range. Perhaps this could be balanced in the introduction by some context around what concentrations are found in the food crops mentioned.
The writing is generally clear, with some minor polishing of language needed.
Figure 4: The scores plots for the PLS-DA model should be omitted, as they have no real meaning. If a metric is needed for their reliability, Cohen's kappa values for predictions of a withheld validation set would be far more meaningful.
In the experimental there needs to be more detail on how the data was split into train, test, and validation sets, as well as any preprocessing that was done on the data. PLS-DA is very prone to overfitting, and these data are wide enough for that to be a major issue.
Also in the experimental, what internal standard was used and how was it used to normalize the data?
Experimental: More details on dimensions of the column, flow rate, and introduction of the sample are needed.
I love the idea of figure 5, but in the PDF file that I was reviewing it is very hard to read, with some bar plots close to illegible. Also, this data would be easier to interpret with boxplots than bar plots.
Section 2.1 title should be revised, as the section is not on how MIC is affected by chlorogenic acid.
What is meant by a decline in MIC values in the sentence "The bacterial cell structure 85 treated with 0.5 MIC CGA exhibited no change and displayed a slight decline in the MIC values"?
Author Response
Point 1: This manuscript details a study on the metabolomic changes in Bacillus subtilis when subjected to high concentrations of chlorogenic acid. The work is scientifically sound, and the statistical analyses were generally well done. The topic is a little underwhelming, as the MIC values reported in the literature and this study are well into the mg/ml range. Perhaps this could be balanced in the introduction by some context around what concentrations are found in the food crops mentioned.
Response 1: Thanks for the comment. CGA can be obtained from many food raw materials (Page 1, Lines 33-35), and its source is safe. We have added the following discussions in the introduction, something like Page 1, Lines 42-44.
Lou et al.[14] reported that the MIC for CGA against B. subtilis was 40 μg/ml. Su et al.[15] indicated that the MIC of CGA against Pseudomonas fluorescein and Staphylococcus saprophytes from chicken was 5 mg/ml.
The bacteriostatic activity of CGA was different, a possible reason is that different CGA extraction conditions and variations in the inoculum level, experimental temperature, and physiological condition of the bacteria used in different studies may affect its antibacterial activity.
Point 2: The writing is generally clear, with some minor polishing of language needed.
Response 2: Thanks for the comment. We have revised the whole manuscript carefully and tried to avoid any grammar or spelling errors.
Point 3: Figure 4: The scores plots for the PLS-DA model should be omitted, as they have no real meaning. If a metric is needed for their reliability, Cohen's kappa values for predictions of a withheld validation set would be far more meaningful.
Response 3: Thanks for the comment. According to your opinions and combined with the content of the article, in order to make the article more concise, we have deleted the scores plots for the PLS-DA model in Figure 4 (Lines 171) and revised the sentence in the text, something like Page 6, Lines 160-161 and Page 7, Lines 172-174.
Point 4: In the experimental there needs to be more detail on how the data was split into train, test, and validation sets, as well as any preprocessing that was done on the data. PLS-DA is very prone to overfitting, and these data are wide enough for that to be a major issue.
Response 4: Thanks for the comment. In this paper, to study the effect of CGA on intracellular metabolites of B. subtilis, three different concentrations were designed and the changes of intracellular metabolites of the bacteria after different time of action were studied. We have added the experimental processing details in the text, something like Page 10-11, Lines 269-280. As follows: “B. subtilis was precultured in LB liquid medium with a rotary speed of 150 rpm at 37 °C for 12 h. Then, the precultured bacteria cells were transferred to a fresh LB liquid medium, and the initial cell density was adjusted to 0.3−0.4 at 600 nm. The cells were cultivated at 37 °C on a rotary shaker at 150 rpm in 250 ml cotton-plugged flasks containing 100 ml of LB liquid medium either with or without CGA. Cell samples were collected at 2, 4, and 8 h either with or without CGA. First, Bacteria cells were harvested by centrifugation at 12000 rpm for 5 min and washed with PBS three times to remove the residual culture medium, followed by washing with ultrapure water to remove the salts from PBS. Then, put the cells on dry ice and add 2 ml of 80 % (v/v) methanol (pre-chilled to -80 °C). Then cells were broken the cells using sonication under 10 °C and incubated the lysate at -80 °C for 2h. Then metabolite-containing were harvested by centrifugation at 12,000 rpm for 10 min at 4-8 °C and collected the metabolite-containing supernatant to a new 1.5-ml tube, and prepared for analyses. Three replicates were performed for each sample.”
Point 5: Also in the experimental, what internal standard was used and how was it used to normalize the data?
Response 5: Thanks for the comment. An in-house database for endogenous metabolites identification was created using Library Manager 2.0 (Thermo Fisher Scientific, CA). Most reference spectra in the internal metabolite library were acquired from chemical standards. In some cases, standard metabolites were not accessible but were observed in biological samples. MS/MS spectra of these compounds were confirmed manually according to Metlin (www.metlin. scripps.edu) or HMDB (www.HMDB.ca) and also saved in Library Manager for reference. All of the areas of acquired peaks were normalized against the internal standard for further data processing.
We have added this explanation in the text, something like Page 11, Lines 295-301.
Point 6: Experimental: More details on dimensions of the column, flow rate, and introduction of the sample are needed.
Response 6: Thanks for the comment. Samples were separated using a synergi Hydro-RP column (2.0×100mm, 2.5 μm, phenomenex). A binary solvent system (mobile phase A, 10 mM tributylamine adjusted with 15 mM acetic acid in water; mobile phase B, methanol) was used. A 25-min gradient with a flow rate of 250 μl/min was applied as follows: 1–5 min at 5% B; 5.1–20 min, 5–90% B; 20.1–25 min, 90% B.
Extraction of Intracellular Metabolites. B. subtilis was precultured in LB liquid medium with a rotary speed of 150 rpm at 37 °C for 12 h. Then, the precultured bacteria cells were transferred to a fresh LB liquid medium, and the initial cell density was adjusted to 0.3−0.4 at 600 nm. The cells were cultivated at 37 °C on a rotary shaker at 150 rpm in 250 ml cotton-plugged flasks containing 100 ml of LB liquid medium either with or without CGA. Cell samples were collected at 2, 4, and 8 h either with or without CGA. First, Bacteria cells were harvested by centrifugation at 12000 rpm for 5 min and washed with PBS three times to remove the residual culture medium, followed by washing with ultrapure water to remove the salts from PBS. Then, put the cells on dry ice and add 2 ml of 80% (v/v) methanol (pre-chilled to -80 °C). Then cells were broken the cells using sonication under 10 °C and incubated the lysate at -80 °C for 2h. Then metabolite-containing were harvested by centrifugation at 12,000 rpm for 10 min at 4-8 °C and collected the metabolite-containing supernatant to a new 1.5-ml tube, and prepared for analyses. Three replicates were performed for each sample.
We have added this details in the text, something like Page 10-11, Lines 269-289.
Point 7: I love the idea of figure 5, but in the PDF file that I was reviewing it is very hard to read, with some bar plots close to illegible. Also, this data would be easier to interpret with boxplots than bar plots.
Response 7: Thanks for the comment. To explain the data more directly and clearly, we have replaced the bar chart with a box chart, something like Page 8, Line 189.
Point 8: Section 2.1 title should be revised, as the section is not on how MIC is affected by chlorogenic acid.
Response 8: Thanks for the comment. We have revised the section 2.1 title, which we have marked with the track changes mode.
Point 9: What is meant by a decline in MIC values in the sentence "The bacterial cell structure 85 treated with 0.5 MIC CGA exhibited no change and displayed a slight decline in the MIC values"?
Response 9: Thanks for the comment. This sentence means to state the damage of different concentrations of CGA on the morphological structure of bacteria. The “decline” means that the integrity of cell morphology and structure is decreased in MIC CGA treatment. In order to more clearly, we have revised the sentence to “The bacterial cells treated with CGA at MIC values were slightly damaged compared with the control and 0.5 MIC CGA cells treated, which had a smooth surface.”
We have revised this sentence in the text, which we have marked with the track changes mode, something like Page 2, Lines 87-90.
